# Reducing the High Iodine Content of *Saccharina latissima* and Improving the Profile of Other Valuable Compounds by Water Blanching

**DOI:** 10.3390/foods9050569

**Published:** 2020-05-04

**Authors:** Cecilie Wirenfeldt Nielsen, Susan Løvstad Holdt, Jens J. Sloth, Gonçalo Silva Marinho, Maren Sæther, Jon Funderud, Turid Rustad

**Affiliations:** 1National Food Institute, Technical University of Denmark, 2800 Kgs. Lyngby, Denmark; suho@food.dtu.dk (S.L.H.); jjsl@food.dtu.dk (J.J.S.); goncalomarinho@hotmail.com (G.S.M.); 2Department of Biotechnology and Food Science, Norwegian University of Science and Technology, 7018 Trondheim, Norway; turid.rustad@ntnu.no; 3Seaweed Energy Solutions AS, 7491 Trondheim, Norway; sather@seaweedenergysolutions.com (M.S.); funderud@seaweedenergysolutions.com (J.F.)

**Keywords:** sugar kelp (*Saccharina latissima*), seaweed, blanching, freezing, iodine, nutrients, bioactives, antioxidant activity

## Abstract

*Saccharina latissima* contains high amounts of iodine in comparison to other seaweeds. The present study aimed to decrease the iodine content of *S. latissima* (sugar kelp) by water blanching and freezing to avoid an excess intake of iodine by consumption of sugar kelp. Various blanching conditions were investigated (temperature; 30, 45, 60 and 80 °C, and duration; 2, 30, 120 and 300 s). Some conditions resulted in a significant decrease in iodine content (≥45 °C and ≥30 s). Non-processed *S. latissima* contained on average 4605 mg iodine kg^−1^ dw^−1^ which significantly decreased following the treatments. The lowest content obtained was 293 mg iodine kg^−1^·dw^−1^ by water blanching at 80 °C for 120 s. The study also investigated if other valuable compounds were affected during the processing conditions. No significant changes were observed for total lipid and protein, but significant changes were seen for ash. A significant loss of two non-essential amino acids (glutamic acid and alanine) due to the blanching process was found. This also resulted in a protein quality increase as the essential amino acid to total amino acid ratio changed from 42.01 ± 0.59% in fresh seaweed to 48.0 ± 1.2% in blanched seaweed. Moreover, the proportion of eicosapentaenoic acid, α-linolenic acid, polyunsaturated fatty acids, and omega-3 fatty acids (%FAME), and the omega-3 to omega-6 fatty acids ratio was significantly higher in the samples blanched at 60 °C for 300 s compared to the fresh and samples blanched at 45 °C for 30 s. The total phenolic content (TPC) and the radical scavenging activity were significantly higher in treated samples. The results indicate that the processing did not compromise the valuable compounds in focus in this study for *S. latissima*; they did, however, result in biomass with an improved profile of health beneficial compounds.

## 1. Introduction

Seaweed as a food source is currently in focus in Europe due to its potential as an environmentally friendly and nutritious food source. It grows in seawater, does not take up any land areas and does not need any freshwater supply. Moreover, seaweeds contain highly valuable bioactive compounds [1], which are of interest from a nutritional point of view. *Saccharina latissima* (Linnaeus) is a brown macroalga commonly known as sugar kelp, which is successfully cultivated in Europe and is commercially available. Valuable compounds specifically for *S. latissima* are minerals, essential amino acids, polyunsaturated fatty acids, phenolic compounds, antioxidants, etc. [2,3,4,5]. Kelps, in general, contain high amounts of the trace element iodine [6] and contents as high as 6500 mg·kg^−1^·dw^−1^ have been found in European sugar kelp [7]. Marine foods are considered rich in iodine and contain up to 30 mg·kg^−1^·dw^−1^ [8]. It is clear that the iodine content of sugar kelp is extremely high.

Due to the possibilities of both insufficient and excess intake of iodine, dietary values for recommended intake (RI) (150 µg iodine day^−1^ for adults) and upper intake level (UL) (600 µg iodine day^−1^ for adults) have been established to provide guidance to consumers by both the Nordic Council of Ministers [9] and the European Food Safety Authorities (EFSA) [10,11]. No maximum levels for iodine in food (including seaweed) have been established in European Food Regulation. However, some member states, e.g., France, have published a recommended maximum level of 2000 mg iodine kg^−1^ dry seaweed product [12].

It is in the manufacturer’s interest, as well as their responsibility, to ensure that their food products are safe and comply with food legislation (Council Directive (EC) 178/2002; [13]). The high iodine content of sugar kelp can, even from low consumption, lead to an intake of iodine above the upper level and hence, this is seen as a possible market barrier for the trade of sugar kelp. Consequently, the European seaweed industry demands procedures that can reduce the iodine content of their final products.

Previous studies have investigated how to reduce iodine in sugar kelp, e.g., by water soaking (low temperatures) or boiling [7,14]. Recently, Stévant et al. (2018) [7] found that soaking in water at 32 °C for 1–6 h would reduce the iodine content significantly by 84%–88%. Another study by Lüning and Mortensen (2015) [14] found a significant iodine reduction of 33% and 75% for sugar kelp boiled for 2 and 20 min, respectively.

The aim of this study was to investigate the possible iodine reduction by processing. Moreover, to investigate if the various processing conditions compromised other valuable compounds, the nutritional value of the final product was determined. The processing conditions investigated were water blanching at short processing times (2, 30, 120 and 300 s) at various water temperatures (30, 45, 60 and 80 °C) or by freezing followed by thawing. Moreover were the protein by sum of amino acids, amino acid profile, total lipid, fatty acid profile, ash, total phenolic content, and antioxidant capacity quantified to evaluate the possible quality compromise of *S. latissima* due to processing. In addition, the true retention factors were calculated in order to show not only the proximate composition, but also link to losses of biomass during processing. Lastly, a brief assessment of the iodine content compared to the recommended intake (RI) and upper intake levels (UL) was conducted.

## 2. Materials and Methods

### 2.1. Chemicals

All chemicals were of analytical grade unless otherwise stated. More specifically, tetra-methyl-ammonium-hydroxide (TMAH) 25%, sodium hydroxide, o-phtaldialdehyde and butylated hydroxytoluene were purchased from Sigma-Aldrich (Steinheim, Germany). Toluene, hydrogen chloride, methanol, chloroform, sodium chloride, and 20% boron trifluoride were purchased from Merck (Darmstadt, Germany). n-Heptane was purchased from VWR (Radnor, Pennsylvania, USA). The purified C23:0 was purchased from Nu Chek Prep (Elysian, MN, USA).

### 2.2. Raw Material

*Saccharina latissima* was harvested at Seaweed Energy Solution’s cultivation site at Frøya, Norway (N63° 42.279’ E8° 52.232′). Approximately 1 m long blades were harvested on 23 April 2018. After harvest, the sugar kelp was stored in aerated tanks with flow-through seawater at 7–8 °C for 2–3 days until processing was carried out. Stem and holdfast were kept, but fouling organisms were removed by hand.

### 2.3. Water Blanching and Freezing

All treatments were performed on samples (*n* = 3) of 150 ± 20.0 g wet weight (*ww*) whole thallus sugar kelp. Water blanching was conducted in a JBN12 (Grant Instruments Ltd., England) water bath in 5 L tap water with the following variables: temperature 30, 45, 60 and 80 °C and processing durations of 2, 30, 120 and 300 s. The water bath was cleaned and the water renewed prior to each treatment process including between treatment replicates. After blanching, the sugar kelp was drained by keeping it vertical for 5–10 s, then placed in a zip lock plastic bag and cooled in ice water for 3 min.

Freezing was conducted in a −20 °C freezing room (Schneider Electric, Rueil-Malmaison, France) for 8 h and thawing was done at 5 °C overnight (freeze-thawed). The freeze-thaw caused drip water to appear, which was drained by keeping it vertical for 5–10 s.

From the harvest batch were three replicates of approximately 150 g fresh sugar kelp collected, drained and stored until analyses. These samples are referred to as “fresh sugar kelp”.

### 2.4. Sample Preparation for Chemical Analysis

Prior to analysis, the sugar kelp samples were cut into 3 × 3 cm pieces and gently mixed. Approximately 20% of the material was used for water and ash analysis. The rest was freeze dried (Alpha 1-4 LDplus, Martin Christ, Germany) at −40 °C, and then homogenized by milling (MM 400, Retsch, Germany) to particle sizes of <300 µm.

### 2.5. Dry Matter and Ash

Dry matter content (DM) was determined gravimetrically by vaporizing water at 105 °C for 20–24 h in an oven (Termaks AS, Bergen, Norway) until stable weight [15]. Ash content was determined gravimetrically by ignition in a muffle furnace (Nabertherm, Lilienthal, Germany) at 600 °C for 15–20 h [16]. Both analyses were performed in duplicates.

### 2.6. Iodine

Inductively Coupled Plasma Mass Spectrometry (ICP-MS) was used for the quantification of the total iodine content in the sugar kelp samples. The samples were prepared according to EN17050:2017 [17]. Briefly, 0.15–0.20 g of freeze-dried milled homogenized powder was weighed into tubes (Kimax^®^). Subsequently, 5 mL Milli-Q^®^ water and 1 mL 25% tetra-methyl-ammonium-hydroxide (TMAH) were added. The tubes were then sealed and placed in a preheated oven at 90 ± 3.0 °C for 3 h followed by cooling and diluting to a final volume of 20 mL with Milli-Q^®^ water. To remove coarse particles, the samples were centrifuged at 10,000× g for 20 min. Prior to analysis, the supernatant was filtered through a 0.45 μm filter and samples were diluted 50 times. Sample extracts were stored in metal free plastic tubes for a maximum of 5 days prior to ICP-MS analysis. The iodine quantification (*n* = 1) was performed by ICP-MS (Finnigan ELEMENT-2, Thermo Fisher, Waltham, MA, USA) combined with an SC2 DX auto sampler and a prepFAST auto dilution system (Elemental Scientific, Omaha, NE, USA). The parameter settings were 15.5 L·min^−1^ coolant gas, 1.1 L·min^−1^ auxiliary gas, and 0.75 L·min^−1^ nebulizer gas. Isotopes monitored were 127I and 185Re for internal standard. The limit of quantification (LOQ) for iodine was 37 μg·g^−1^. The certified reference material (NIST 3232, Kelp powder) was analyzed together with the samples and the obtained results complied well with the certified value (recovery 96.8%, *n* = 2).

### 2.7. Amino Acid Hydrolysis and Calculation of Protein Content

Briefly, 50 mg sample was hydrolyzed in 1 mL 6 M HCl at 105 °C for 22 h (*n* = 2). Prior to quantification, the samples were neutralized by NaOH and HCl to pH 7.0 ± 0.5 and filtered through a Whatman glass microfiber filter (GF/C) using suction. The samples were diluted 1:100 with distilled water followed by a 0.22 μm filtration. Then the amino acids were quantified by High-Performance Liquid Chromatography (HPLC) (Dionex UltiMate^®^ 3000 HPLC+ focused, Dionex UltiMate^®^ 3000 Autosampler, Dionex RF Fluorescence Detector, Thermo Scientific, USA) including precolumn derivatization of the amino acids with o-phtaldialdehyde and Nova-Pak^®^ column (C18, 4 μm). Tryptophan was destroyed in acid hydrolysis, thus not detected. The chromatographic peaks for glycine and arginine gathered in one, therefore an average of their molar masses was used to calculate their content.

The total protein content was calculated by summing the total moles of amino acids as recommended by Angell et al. (2016) [18] and FAO (2003) [19] and then subtracting the water mass (18 g H_2_O mol^−1^ amino acid), which was integrated during disruption of peptide bonds in the acid hydrolysis [20].

### 2.8. Determination of Total Lipid Content

The gravimetric method described by Bligh and Dyer (1959) [21] was used to quantify total lipid content. Briefly, a mixture of demineralized water, methanol and chloroform (0.8:2:1 mL) was added to 30 mg freeze-dried sample followed by homogenizing with 1 mL chloroform (20 s) and then 1 mL demineralized water (20 s). The mixture was centrifuged thoroughly at 4 °C. The chloroform phase (0.5 mL) was added to a pre-weighed glass container and vaporized overnight in a fume hood. The following day the container was weighed again. The total lipid content was calculated by the following Equation:(1)$$\mathrm{tatal}\;\mathrm{lipid}\left(\%\right)=\frac{1v\cdot C\cdot 100}{\mathrm{Cv}\cdot m}$$
where l_v_ is the lipid weighed after vaporization in mg, c is the added chloroform (2 mL), c_v_ is the vaporized chloroform (0.5 mL), and m is the mass of the weighed sample before extraction.

### 2.9. Carbohydrates by Difference

Estimation of the total carbohydrate content was done by “total carbohydrate by difference” [19], which includes fibers:(2)carbohydrates=100−(weightinggram [protein+lipid+water ash]in 100 g of food)

### 2.10. Fatty Acids

Direct methylation of fatty acids were performed according to [22]. Approximately, 100 mg of sample was mixed with 1 mL 1.0 M NaOH in methanol, 1 mL toluene, and 0.1 mL 2% (*w*/*v*) C23:0 in *n*-heptane and sonicated for 10 min, followed by 100 °C water bath for 2 min and cooling in cold water. Next, 2 mL boron trifluoride (20% solution) were added and boiled and cooled as earlier described. Lastly, 2 mL 6.8 M saturated sodium chloride solution along with 1 mL heptane with 0.01% butylated hydroxytoluene were added and shaken. The heptane phase was transferred to a GC vial and FAMEs were analyzed by GC (HP 5890A, Agilent Technologies, Palo Alto, CA, USA) according to the American Oil Chemists’ Society (AOCS) [23]. For separation, DB127-7012 column (10 m × ID 0.1 mm × 0.1 µm film thickness, Agilent Technologies, Palo Alto, CA, USA) was used. Injection volume was 0.2 µL in split mode (1:50). The initial temperature of the GC-oven was 160 °C. The temperature was set to increase gradually as follows: 160–200 °C (10.6 °C min^−1^), 200 °C kept for 0.3 min, 200–220 °C (10.6 °C min^−1^), 220 °C kept for 1 min, 220–240 °C (10.6 °C min^−1^) and kept at 240 °C for 3.8 min. The determination was conducted in duplicates. Fatty acids were identified by comparison of retention times with those from a mixture of known fatty acid standards. Results were given in area %.

### 2.11. Extraction for Antioxidant Analyses and Total Phenolic Content

The extraction of antioxidants was executed according to [24] with some modifications. Briefly, 0.2 g sugar kelp sample was weighed, and 5 mL methanol was added, and the samples were placed in a sonicator for 30 min. Thereafter, the samples were centrifuged (2164× g for 10 min) and the supernatant was collected. The pellets were resuspended and extractions repeated twice. The solvent was evaporated under nitrogen flow. When the extracts were completely dry, they were stored in the freezer (−18 °C). Prior to the analyses, the dried powders were dissolved in 1 mL methanol.

### 2.12. Total Phenolic Content (TPC)

The procedure was carried out according to [25]. Extracts (100 µL) were mixed with Folin–Ciocalteu reagent (0.75 mL, 10% *v*/*v*). After 5 min, 0.75 mL sodium carbonate (7.5% *w*/*v*) was added to the mixture, which was then incubated at room temperature in darkness for 90 min. The absorbance was measured at 725 nm. The measured absorbance was converted into gallic acid equivalents by a standard curve of gallic acid in the range of 7.8–250 µg mL^−1^.

### 2.13. DPPH Radical Scavenging Activity

The radical scavenging activity was performed according to the method described by Yang, Guo and Yuan (2008) [26] with modifications. Briefly, 100 µL of methanolic extracts were added to a microplate followed by 100 µL 0.1 mM DPPH soluted in methanol and mixed followed by incubation for 30 min in darkness at room temperature. The absorbance was measured at 517 nm in a microplate reader (Synergy 2 BioTek, Winooski, VT, USA). Triplicate measurements were performed and butylated hydroxytoluene (BHT) was included in the assay as a positive control since a concentration of 0.91 mM of BHT is giving approximately 70% inhibition. A sample blank was made with DPPH but without extract solution (Ab) and a sample control was made without DPPH but with extract/fraction solution (A0). Results are expressed as IC50, i.e., the concentration of extract needed to obtain 50% inhibition. The % DPPH radical scavenging activity was calculated as follows:(3)$$\mathrm{DPPH}\;\mathrm{radical}\;\mathrm{scavenging}\;\mathrm{activity}=\left(1-\frac{\mathrm{As}-A0}{\mathrm{Ab}}\right)\times 100$$

### 2.14. Mass Balances and True Retentions

All samples were weighed before and after treatment with one decimal accuracy. Before weighing, the samples were drained by keeping them vertical for 5–10 s. The true retention (TR) of a compound is the proportion of a particular nutrient that remains after processing relative to the original content of that specific nutrient. True retentions were calculated based on the proximate composition before and after processing and were calculated as suggested by [27]:(4)$$\mathrm{TR}=\frac{g\;\mathrm{retasined}\;\mathrm{nutrient}\;\cdot g\;\mathrm{total}\;\mathrm{product}\;\mathrm{post}\;\mathrm{treatment}}{g\;\mathrm{original}\;\mathrm{nutrient}\;\cdot g\;\mathrm{total}\;\mathrm{product}\;\mathrm{prior}\;\mathrm{treatment}}$$

In cases where a replicate of a specific nutrient concentration was missing due to analytical mistakes, the missing replicate was interpolated from the other analytical replicates by taking an average of the known replicates for that specific treatment.

### 2.15. Statistical Analysis

The results are given as mean ± standard deviation. The statistical analyses were carried out in the software SPSS Statistics 24 (IBM Corp., Armonk, NY, USA). The test run to define the statistically significant difference between the means of the groups (fresh, freeze-thawed, and blanched material) was a one-way ANOVA with Tukey’s post-hoc test. A one-way PERMANOVA was used to test the effect of processing on total phenolic content, and radical scavenging activity (PERMANOVA package in PRIMER+; [28]; type III sum of squares and unrestricted permutation (9999) on raw data; α = 0.05) with a posteriori analysis (pairwise test). Means were considered statistically significantly different when levels of *p* < 0.05 were obtained 

## 3. Results and Discussion

### 3.1. Iodine Content of Sugar Kelp

Fresh Norwegian sugar kelp harvested in April 2018 contained 4605 ± 274 mg iodine·kg^−1^·dw^−1^, which is comparable to other European cultivated sugar kelp (3460–6568 mg·kg^−1^·dw^−1^) [6,7,29,30]. The process of freeze-thawing sugar kelp did not decrease the iodine content significantly (one-way ANOVA; F = 117, df = 15, *p* < 0.001) (Table 1). However, water blanching decreased the iodine content significantly for all blanching treatments except when treated at 30 °C for 2 s. All blanching treatments, except 30 °C below 120 s and 45 °C at 2 s, sufficiently reduced the iodine content below the maximum level of 2000 mg·kg^−1^·dw^−1^ as recommended by ANSES (2018) [12] in seaweed products. The iodine content in the blanched sugar kelp approached a constant level for various treatments with an average content of 328 ± 19 mg·kg^−1^·dw^−1^ (Figure 1). Similarly, Stévant et al., (2018) [7] also reported that a constant level was achieved when subjecting *S. latissima* to warm water at 32 °C for 1 h. The most efficient treatment in this present study reduced the iodine content to 12% relative to the initial iodine content in fresh sugar kelp.

For a better perspective of the iodine content and safe intake of sugar kelp, the recommended intake (RI) and upper intake levels (UL) for adults are used [9,10,11]. If considering the only dietary source of iodine for an adult was from sugar kelp, then to reach the RI and UL 0.35 g or 1.4 g of fresh non-processed sugar kelp should be consumed, respectively. In the case of blanched sugar kelp then 9.2 or 37 g of sugar kelp could be consumed for RI and UL, respectively. A risk assessment considering other sources of iodine in a daily diet should be taken into consideration when evaluating the potential risk of sugar kelp consumption, but overall, this study proves that a reduction of iodine in sugar kelp can be obtained.

The iodine content reached a constant level at 120 s for the treatments at 45 °C and 60 °C, thus the treatments at those temperatures with a longer processing time (300 s) did not undergo further chemical analysis. The 30 °C treatments and the freeze-thawed treatments were also not further investigated, as the treatments did not reduce the iodine content as satisfactorily as the others.

### 3.2. Proximate Composition

The proximate composition (*n* = 3) for the selected treatments can be found in Table 2. Ash content of fresh sugar kelp was 44.51 ± 0.86% dw. The content of ash varied significantly for all water-blanched samples (one-way ANOVA; F = 79, df = 10, *p* < 0.001). The ash content for water-blanched samples was between 9.1 ± 1.6% dw and 26.3 ± 1.5% dw. Protein content in fresh sugar kelp was 7.9 ± 2.5% dw, and in the blanched samples it was 9.8 ± 3.0% dw to 15.3 ± 2.6% dw. No significant differences were found for protein content between any of the samples (one-way ANOVA; F = 2.4, df = 10, *p* = 0.064). The lipid content was 5.8 ± 2.6% dw in fresh *S. latissima* and varied for the blanched samples between 6.9 ± 0.8% dw to 10.2 ± 0.6% dw with no significant differences (one-way ANOVA; F = 1.8, df = 10, *p* = 0.132). Carbohydrates were calculated from the other proximates. As the ash content showed significant difference between treatments, the carbohydrates also showed significant variations (one-way ANOVA; F = 14, df = 10, *p* < 0.001).

### 3.3. Retention of Nutrients

The proximate composition is given for samples after each individual treatment, but does not take the potential loss of biomass into consideration due to processing. From the proximate composition, it cannot be concluded that there is a loss or gain due to treatment, therefore the true retention factors were calculated relative to fresh sugar kelp (Table 3). In Figure 2 the concentrations of each individual proximate and the true retention factors are seen. This together defines the mass balances of the treatments, which can indicate if there is a loss of each individual proximate. The total loss of ash, protein, lipid, and carbohydrate all together (total proximate) for each individual treatment are given relative to the fresh *S. latissima*. Fresh sugar kelp had a total proximate composition of 9.3 g 100 g^−1^ ww and the retained total amount of each treatment varied from 4.3 to 6.6 g 100 g^−1^ ww. The treatment that had the least loss (29%) of proximate was 45 °C at 2 s, whereas the others had a loss that ranged from 41% to 54%.

To the best of the authors’ knowledge, no earlier study has explored the potential loss of proximate composition in *S. latissima,* due to water blanching. Therefore, comparisons to peer-reviewed studies on vegetables were performed. Water blanching (60, 150 and 180 s) of bell peppers, peas, and potatoes led to a significant protein loss of between 8% and 24% [31,32]. In this current study a significant difference in protein was only found for 60 °C at 30 s with a retention of 0.48 ± 0.16 (Tukey’s post-hoc test; *p* = 0.047). In all other cases, no significant differences were found due to high standard deviations (Tukey’s post-hoc test; *p* > 0.154). The true retention factor of protein was on average 0.67, meaning a total protein loss of 33%, which was higher than that found for vegetables. The high standard deviations are most likely due to the method of treating the product after blanching. The sugar kelp surface consists of mucus and seawater, which was probably interfering with the blanching water during treatment. If the mucus and seawater were washed away during blanching it would be replaced by blanching water on the surface. By shaking the sugar kelp consistently, it was expected that the blanching water would be removed, but some of the blanching water would stay on the product surface and interfere when weighing the samples, creating high standard deviations. The true retention factors and the mass balances indicated a significant loss for the ash content (one-way ANOVA; F = 297, df = 10, *p* < 0.001). *Saccharina latissima* is rich in minerals and trace elements such as Na, K, Mg, and Fe [33]. This significant loss of ash is probably not only due to the loss of iodine, but also other minerals and trace elements. If these minerals are located on the surface of the sugar kelp, they could dissolve into the water when blanched. Moreover, the relatively high processing temperature and low salinity of the blanching water could create a shock to the cells, leading to cell bursts and protein and minerals leaking from the cells.

### 3.4. Amino Acid Composition

The amino acid composition was quantified for selected water blanching treatments and the fresh sugar kelp sample. Two of the amino acids (glutamic acid and alanine) had a significant loss due to treatments when compared to the fresh sugar kelp (Figure 3). Fresh sugar kelp contained significantly higher amounts of glutamic acid (173 mg·g^−1^ protein; one-way ANOVA; F = 6.4, df = 10, *p* < 0.001) when compared to the treated samples, although not when compared to the 45 °C at 2 and 30 s (Tukey’s post-hoc test; *p* > 0.395). This meant that the treatments with higher temperatures and process times had a significant loss of glutamic acid. On average, the significantly different samples had an average content of 146 mg glutamic acid g^−1^ protein. For alanine, the fresh sample was significantly different to the treated samples, meaning that there was a significant loss due to processing (one-way ANOVA; F = 52, df = 10, *p* < 0.001). The content in fresh sugar kelp was 178 mg alanine·g^−1^ protein and the average of the treated samples were 128 mg alanine·g^−1^ protein. The entire amino acid profile can be found in the data repository.

No significant changes were found for the essential amino acids. The essential amino acid to total amino acid ratio (EAA ratio) increased, since there was a significant loss of the non-essential alanine and glutamic acid. Fresh sugar kelp had an EAA ratio of 42.01 ± 0.59% EAA, and this was comparable to studies from Denmark and the Faroe Islands [2,34]. Whereas, the blanched samples on average had a ratio of 48.0 ± 1.2%, and were significantly different from the fresh sample (one-way ANOVA; F = 7.9, df = 10, *p* < 0.001).

The limiting amino acid for all samples was histidine, which is also seen in other studies [2,34]. The amino acid score (not considering digestion) was on average above 100% (108 ± 12%), with no significant differences between neither sample (one-way ANOVA; F = 1.1, df = 10, *p* = 0.388). Summing up, the blanching treatment did not compromise the amino acid quality but actually increased it as two non-essential amino acids had a significant loss.

### 3.5. Fatty Acid Composition

The fatty acid (FA) profile was quantified by direct methylation and given in % FAME for fresh sugar kelp and the samples blanched at 45 °C and 60 °C for 30 s and 300 s, respectively. The complete FA profile (% FAME) can be found in the data repository. The quality of FAs can be explained by the content of the individual fatty acids, which the human body cannot synthesize—α-linolenic acid (ALA) and linoleic acid (LA), but also the two fatty acids docosahexaenoic acid (DHA) and eicosapentaenoic acid (EPA). Moreover, the total amount of polyunsaturated fatty acids (PUFA) and omega-3 fatty acids (n-3) as well as the ratio of n-3 FA to omega-6 fatty acids (n-6) indicate the quality of the lipid fraction. The 60 °C 300 s blanched samples presented a higher proportion of EPA, ALA, PUFA, and n-3, and a higher n-3/n-6 ratio compared to the fresh and 45 °C 30 s blanched samples (Table 4). The increased proportion results from the reduction of other fatty acids, namely unsaturated and monounsaturated, during the processing. Overall, this results in a biomass with an improved profile of health-beneficial fatty acids. No significant difference was found for LA, while DHA, which was present in the sugar kelp in very low amounts, seem to be degraded during processing.

### 3.6. Antioxidant Activity and Total Phenolic Content

Processing had a significant effect on the amount of methanolic extract obtained (one-way PERMANOVA; F = 20.5, df = 3, *p* = 0.002), with significantly higher amounts extracted from the fresh samples compared to the water-blanched samples (7.2–11.1% dw; Figure 4). This demonstrates that blanching sugar kelp will result in a significant amount of compounds being transferred to the water phase or degraded during processing.

Blanching had a significant effect on total phenolic content (TPC) (one-way PERMANOVA; F = 26.0, df = 3, *p* = 0.0011; Figure 5). TPC was higher in the 60 °C 300 s treated sample (*p* = 0.013) compared to the fresh sample. On the other hand, there was no significant difference in TPC between fresh and 45 °C 30 s blanched samples (*p* = 0.07). TPC found in the present study for fresh is within the range of the values reported for *S. latissima* harvested at different seasons (0.84–2.41 mg·GAE/g sugar kelp [5]). On the other hand, TPC values obtained for the blanched samples are above those reported in the same study.

TPC results expressed in gallic acid equivalents per mg of extract revealed an even greater effect of processing on the TPC. Blanching significantly increased the content of TPC as compared to the fresh sugar kelp (one-way PERMANOVA; F = 392, df = 3, *p* < 0.01, Figure 6). The highest TPC was found in the 60 °C 300 s blanched samples (*p* < 0.013), followed by the 45 °C 30 s blanched samples, and then fresh samples (*p* = 0.96). These results suggest that the extraction of other compounds during water blanching may have resulted in concentration of phenolic compounds in the processed sugar kelp.

DPPH radical scavenging activity revealed a concentration dependency and increased with increasing concentrations of algal extract (data not shown). Processing increased the radical scavenging activity significantly (F = 13.5, df = 3, *p* = 0.0053, Figure 7). These results suggest that compounds with high radical scavenging activity are retained and up-concentrated in the sugar kelp during water blanching. This correlates well with the current results; TPC have been identified as a major component contributing to radical scavenging activity of seaweed [5,35,36,37,38].

## 4. Conclusions

This study showed that water blanching is a promising approach for reducing the iodine content in Norwegian-cultivated *Saccharina latissima*. Up to 88% reduction was obtained by blanching at optimized conditions (≥45 °C and ≥30 s. Considering the recommended intake and upper intake level reported by the Nordic Nutrition Recommendations (2012)). If sugar kelp was the only source of dietary iodine, a maximum 9.2 g or 37 g, respectively of blanched sugar kelp can be consumed daily to avoid exceeding these recommendations. However, freeze-thawing did not decrease the iodine content of sugar kelp. These are important findings for the food-producing industry that is using seaweed as a raw material and is responsible for consumer safety. In terms of processing effects on other nutritionally valuable compounds, the treatment that had the least loss (29%) of total proximate composition was 45 °C at 2 s, whereas the other treatments had a loss that ranged from 41% to 54%. More specifically, a significant loss of ash occurred, which is comparable with the degree of loss of iodine together with other minerals. Water blanching also caused a significant loss of two amino acids (glutamic acid and alanine), which led to a higher EAA/AA ratio. Moreover, water blanching resulted in biomass with an improved composition of health beneficial compounds, namely PUFA and phenolic compounds, and antioxidant activity.

In perspective, other valuable compounds with antioxidant activity found in sugar kelp such as the pigment fucoxanthin and carbohydrates could have been interesting to study. Moreover, the change in texture, color, and taste (e.g., umami) due to blanching is also interesting and worth further study.

## Figures and Tables

**Figure 1 foods-09-00569-f001:**
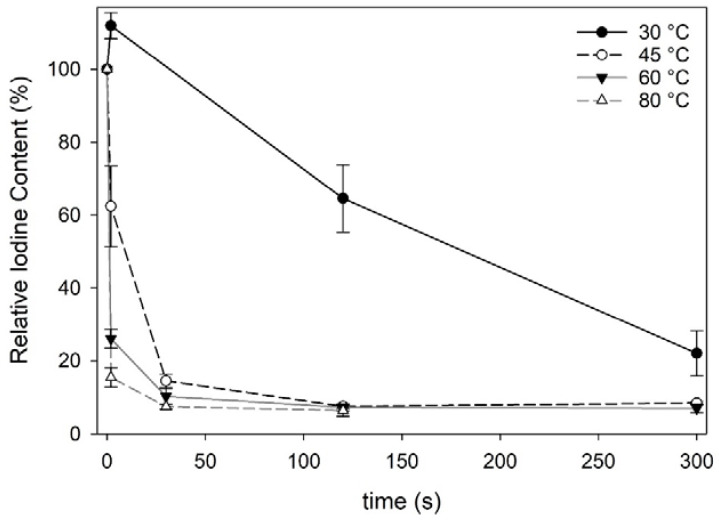
Iodine content in water blanched *Saccharina latissima* relative to fresh *S latissima* expressed in %. Each data point represent the mean iodine content with standard deviations (*n* = 3).

**Figure 2 foods-09-00569-f002:**
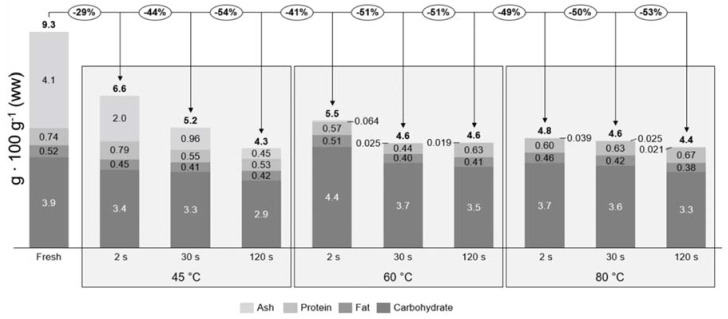
Mass balances for the proximate composition relative to the fresh sugar kelp for each blanching treatment. The percentages in circles describe the total loss of the proximate composition (excluding water). The concentration of each proximate is described by the bar diagram and the bold number above the bars are the total proximate composition relative to wet weight.

**Figure 3 foods-09-00569-f003:**
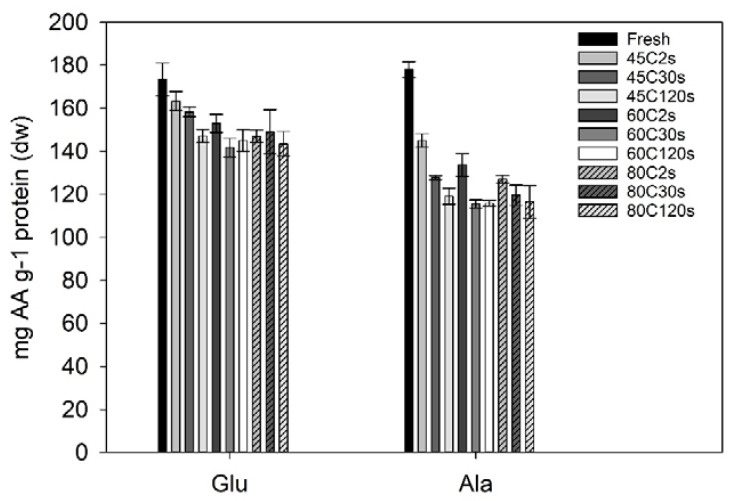
Glutamic acid and alanine in *Saccharina latissima* after different treatments given in mg amino acid (AA) g^–1^ of protein. Error bars represent the standard deviation (*n* = 3). A one-way ANOVA indicated a significant difference between the fresh sugar kelp sample compared to the treated sugar kelp for both glutamic acid (Glu) and alanine (Ala).

**Figure 4 foods-09-00569-f004:**
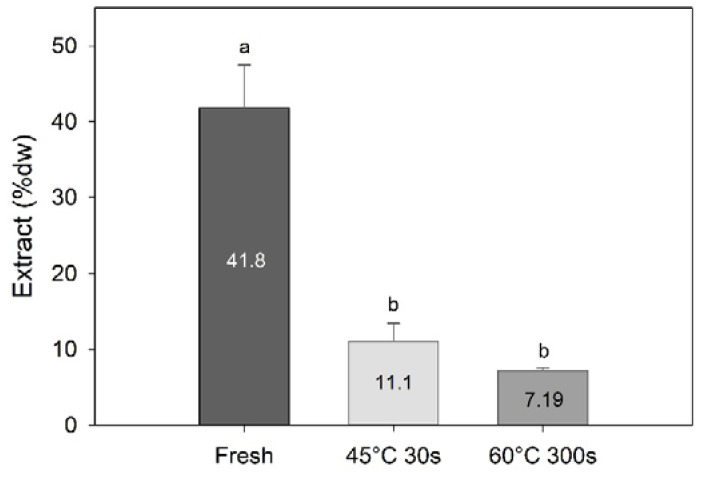
Amount of methanolic extracts (% dw) of sugar kelp for fresh and two different blanching treatments. Different letters represent a significant difference (*p* < 0.05) between treatments. Data are mean ± SD; *n* = 3.

**Figure 5 foods-09-00569-f005:**
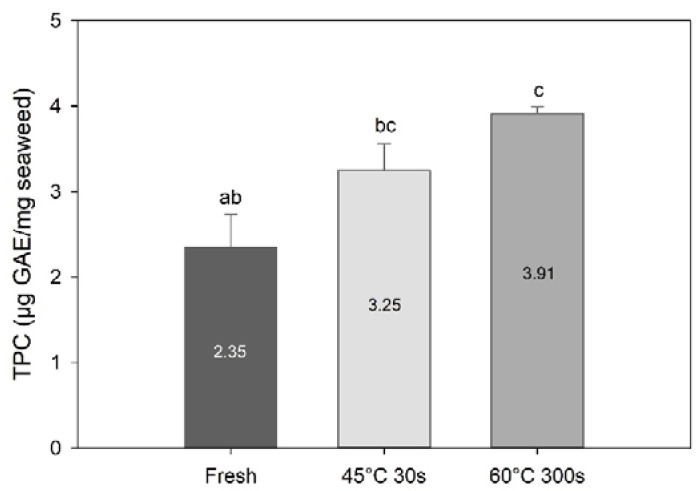
Total phenolic content of fresh and selected blanched samples of *Saccharina latissima* expressed in gallic acid equivalents per mg of freeze-dried samples. Data are mean ± SD; *n* = 3. Different letters represent a significant difference between treatments (*p* < 0.05).

**Figure 6 foods-09-00569-f006:**
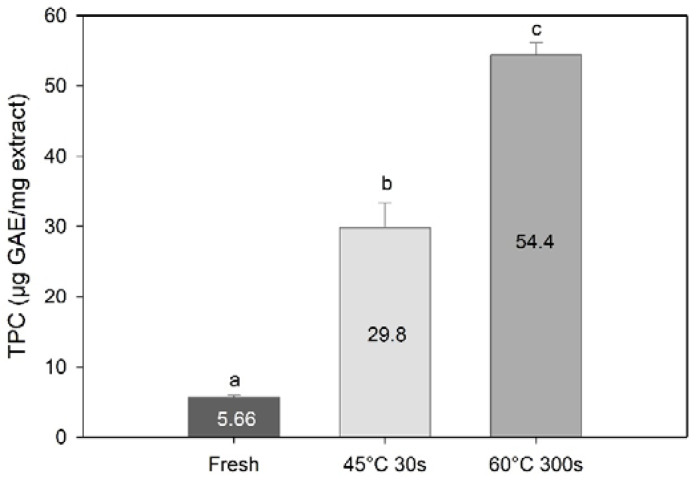
Total phenolic content expressed in gallic acid equivalents per mg of methanolic extract of *Saccharina latissima* from fresh and two types of blanching. Data are mean ± SD; *n* = 3. Different letters represent a significant difference (*p* < 0.05) between treatments.

**Figure 7 foods-09-00569-f007:**
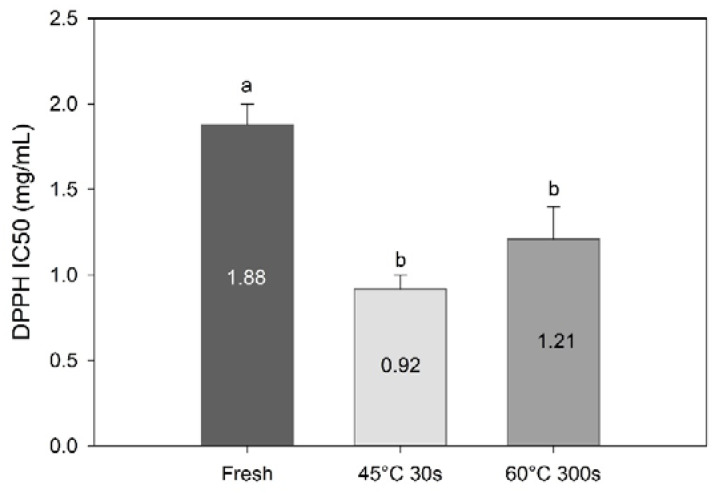
DPPH radical scavenging activity (IC50; mg/mL) of methanolic extracts of fresh and blanched *Saccharina latissima*. Different letters represent a significant difference (*p* < 0.05) between treatments. Data are mean ± SD; *n* = 3.

**Table 1 foods-09-00569-t001:** Iodine content in fresh, freeze-thawed and water-blanched *Saccharina latissima* expressed in mg·kg^−1^·dw^−1^. Results are mean ± standard deviation (*n* = 3).

Time	Temperature/Treatment	Iodine (mg·kg^−1^·dw^−1^)
N/A	Fresh	4605 ± 274 ^ab^
N/A	Freeze-thawed	4057 ± 419 ^b^
2 s	30 °C	5157 ± 201 ^a^
	45 °C	2873 ± 627 ^c^
	60 °C	1198 ± 146 ^d^
	80 °C	711 ± 151 ^de^
30 s	45 °C	667 ± 120 ^de^*
	60 °C	472 ± 121 ^de^
	80 °C	343 ± 41 ^e^
120 s	30 °C	2973 ± 523 ^c^
	45 °C	346 ± 35 ^e^
	60 °C	334 ± 55 ^e^
	80 °C	293 ± 90 ^e^
300 s	30 °C	1014 ± 349 ^de^
	45 °C	388 ± 23 ^de^
	60 °C	321 ± 68 ^e^

N/A designates not applicable. (*) indicates two replicates (*n* = 2). Letters (a–e) denote significant differences between treatments by one-way ANOVA and Tukey’s post-hoc test.

**Table 2 foods-09-00569-t002:** Proximate composition of fresh and water-blanched *Saccharina latissima.* Data are expressed as means ± SD and represent three process replications (*n* = 3). Water is given in % ww, whereas ash, protein (total amino acids), fat, and calculated carbohydrates are given in % dw.

Component	Fresh			45 °C			60 °C			80 °C
		2 s	30 s	120 s	2 s	30 s	120 s	2 s	30 s	120 s
Water	90.68 ± 0.30 ^a^	93.42 ± 0.77 ^b^	94.79 ± 0.47 ^c^	95.70 ± 0.20 ^c^	94.49 ± 0.46 ^bc^	95.45 ± 0.20 ^c^	95.44 ± 0.28 ^c^	95.23 ± 0.20 ^c^	95.36 ± 0.03 *^c^	95.64 ± 0.14 ^c^
Ash	44.51 ± 0.86 ^a^	26.3 ± 1.5 ^b^	18.4 ± 1.7 ^cd^	10.8 ± 2.5 ^ef^	20.5 ± 3.2 ^bc^	12.3 ± 2.8 ^def^	9.1 ± 1.6 ^f^	17.2 ± 1.7 ^cde^	11.7 ± 1.2 *^def^	11.2 ± 1.4 ^def^
Protein	7.9 ± 2.5 ^a^	11.8 ± 2.4 ^a^	10.5 ± 1.4 *^a^	12.3 ± 1.0 ^a^	10.2 ± 3.0^a^	9.8 ± 3.0 ^a^	13.6 ± 1.8 ^a^	12.6 ± 2.3 ^a^	13.6 ± 2.3 ^a^	15.3 ± 2.6 ^a^
Fat	5.8 ± 2.6 ^a^	6.9 ± 0.8 ^a^	7.9 **	10.2 ± 0.6 *^a^	9.1 ± 1.5^a^	8.6 ± 4.0 *^a^	9.0 ± 1.7 ^a^	9.7 ± 0.7 *^a^	9.1 ± 1.5 ^a^	8.7 ± 1.3 *^a^
Carbohydrates	41.8 ± 4.7 ^a^	55.0 ± 0.3 ^abc^	65.2 **	68.7 ± 3.9 ^cd^	60.1 ± 5.0^bcd^	65.3 ± 4.6 *^cd^	68.3 ± 1.4 ^cd^	60.2 ± 1.8 *^cd^	64.9 ± 0.6 *^cd^	63.7 ± 1.4 *^cd^

(*) Included only duplicates (*n* = 2), (**) indicated one replicate (*n* = 1). Means with different letters (a–f) within each row are significantly different (*p* ≤ 0.05).

**Table 3 foods-09-00569-t003:** True retention factors post processing relative to the fresh sugar kelp. The retention factors are presented in means ± SD (*n* = 3).

Component			45 °C			60 °C			80 °C
	2 s	30 s	120 s	2 s	30 s	120 s	2 s	30 s	120 s
Water	0.87 ± 0.03 ^a^	0.99 ± 0.06 ^a^	0.93 ± 0.13 ^a^	0.83 ± 0.12 ^a^	0.83 ± 0.04 ^a^	0.87 ± 0.03 ^a^	0.74 ± 0.05 ^a^	0.81 ± 0.04 ^a^	0.86 ± 0.13 ^a^
Ash	0.39 ± 0.07 ^b^	0.22 ± 0.02 ^c^	0.09 ± 0.02 ^d^	0.013 ± 0.005 ^d^	0.005 ± 0.001 ^d^	0.004 ± 0.000 ^d^	0.007 ± 0.001 ^d^	0.005 ± 0.001 ^d^	0.004 ± 0.001 ^d^
Protein	0.89 ± 0.22 ^ab^	0.69 ± 0.05 ^ab^	0.63 ± 0.11 ^ab^	0.65 ± 0.28 ^ab^	0.48 ± 0.16 ^b^	0.70 ± 0.11 ^ab^	0.57 ± 0.07 ^ab^	0.67 ± 0.11 ^ab^	0.73 ± 0.09 ^ab^
Fat	0.73 ± 0.11 ^a^	0.74 ± 0.06 ^a^	0.69 ± 0.07 ^a^	0.79 ± 0.23 ^a^	0.60 ± 0.26 ^a^	0.79 ± 0.23 ^a^	0.60 ± 0.26 ^a^	0.65 ± 0.14 ^a^	0.62 ± 0.03 ^a^
Carbohydrates	0.72 ± 0.12 ^ab^	0.79 ± 0.10 ^ab^	0.65 ± 0.08 ^b^	0.89 ± 0.16 ^ab^	0.74 ± 0.01 ^ab^	0.74 ± 0.01 ^ab^	0.66 ± 0.08 ^b^	0.71 ± 0.04 ^ab^	0.69 ± 0.10 ^ab^

Means with different letters (a–f) within each row are significantly different (*p* ≤ 0.05). The factors are relative to the fresh sugar kelp, which had a factor of 1.0 and statistical letter (a). Statistical descriptions: water (one-way ANOVA; F = 3.6, df = 10, *p* = 0.006), ash (one-way ANOVA; F = 297, df = 10, *p* < 0.001), protein (one-way ANOVA; F = 2.2, df = 10, *p* = 0.060), fat (one-way ANOVA; F = 1.3, df = 10, *p* = 0.287), and carbohydrates (one-way ANOVA; F = 2.7, df = 10, *p* = 0.025).

**Table 4 foods-09-00569-t004:** Fatty acid composition of fresh and blanched sugar kelp expressed in % FAME. Data are expressed as means ± SD and represents three process replications (*n* = 3). The fatty acids are given in % FAME although the ratio (n-3/n-6) is without unit.

Fatty Acids	Fresh	45 °C	60 °C
		30 s	300 s
18:2 (n-6) (LA)	4.96 ± 0.12 ^ab^	5.50 ± 0.23 ^a^	4.87 ± 0.16 ^b^
18:3 (n-3) (ALA)	15.2 ± 1.5 ^a^	18.1 ± 2.1 ^ab^	22.63 ± 0.45 ^b^
20:5 (n-3) (EPA)	12.18 ± 0.82 ^a^	13.2 ± 1.0 ^a^	17.38 ± 0.16 ^b^
22:6 (n-3) (DHA)	0.36 ± 0.02 ^a^	0.15 ± 0.08 ^b^	0.00 ± 0.00 ^c^
n-3	29.0 ± 2.4 ^a^	32.2 ± 3.0 ^a^	41.20 ± 0.71 ^b^
n-6	22.51 ± 0.91 ^a^	25.37 ± 0.61 ^b^	26.00 ± 0.03 ^b^
n-3/n-6	1.29 ± 0.08 ^a^	1.27 ± 0.09 ^a^	1.59 ± 0.03 ^b^
PUFA	51.5 ± 3.1 ^a^	57.6 ± 3.6 ^a^	67.19 ± 0.67 ^b^

(a–e) denote significant difference between sample treatments. From the top linoleicacid (LA), α-linolenic acid (ALA), eicosapentaenoic acid (EPA), docosahexaenoic acid (DHA), total omega-3 fatty acids (n-3), total omega-6 fatty acids (n-6), the omega-3 to omega-6 fatty acids ratio (n-3/n-6), and total polyunsaturated fatty acids (PUFA).

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
