# Peer review of "Reducing the High Iodine Content of Saccharina latissima and Improving the Profile of Other Valuable Compounds by Water Blanching"

_foods, 2020, doi:10.3390/foods9050569_

Round 1
Reviewer 1 Report
The manuscript is focused on the applicability of different treatments (blanching, freezing) to reduce the iodine content of Saccharina Latissima. Another claimed drawback is In general, the study is not particularly innovative, but the approach and the methodology are peculiar and well-applied. The paper is focused on an attractive research topic; experimental design is well organized and statistical approach is rigorous.
In my opinion, language and sentence clarity should be improved in the whole manuscript, as some parts of the manuscript, were difficult to follow.
Also, more attention should be given to manuscript formatting, as also suggested in some specific comments reported below. Please consider to revise the whole manuscript accordingly
L23: please, remove the double point.
L40: Please, correct the citation or remove it.
L63-65: I would suggest rephrasing this sentence, as it is not clear.
L85-86: From an industrial point of view, would it be practical to seal the seaweed in plastic bath after heat treatment, prior to cooling? I would expect a continuous or semi continuous process, without the need of any intermediate packaging step. If it is true, why did you choose to pack the blanched seaweed?
L89-90: If I understood well, fresh seaweed was analyzed for every batch and were considered as the control. Please clarify this concept.
L103: I think that the sentence is missing a part.
L115: The heading should be modified according to the purpose of these analyses.
L118: Please, try to clarify these steps. This sentence is not clear.
L122: Please, remove the paragraph.
L125: missing verb.
L130: It would be useful to briefly describe the reported method.
Equation 3: No information has been reported about the seaweed weighting step.
L180: What do you mean with data extrapolation? Please explain and report it.
L199, elsewhere: Please, format citations according to reference guidelines
Table 3: Why for fresh seaweed retention factors were reported? As reported by equation 3, TR is defined according to the losses of component subsequent processing. I can not understand how the fresh non processed seaweed can be affected. If it was no processed.
L278: please add relatively before high
Reviewer 2 Report
Given the increased request and role of seaweeds as novel food in the continental diet, the conclusions of this study are of interest for the food processing sector and regulations. The manuscript is generally well written and the conclusion are supported by the data presented. I only have a few minor spelling points:
Line 23: delete the extra dot.
Lines 388-390: check tenses and construction of the phrase.
Reviewer 3 Report
The manuscript with reference ID foods-773034 evaluated the effectiveness of a heat treatment in reducing the high iodine content of Saccharina Latissima combined with an improvement of another valuable compounds”. This is a valuable study presenting the effects of two treatments for reduction of the iodine content of one seaweed edible that intends to increase its inclusion in the human diet and for that the level of this compounds should be evaluated for health security reasons. Also, another beneficial compounds with relevant impact on nutritional and health consumer richness have been investigated. Below, the authors can see my suggestions, comments and observations that enhance the quality of manuscript.
Always that the authors indicating a significant effect of blanching treatment in a specific compound, addition of P-value is essential.
Abstract:
Page1, Line 15 – change the “---other foods…” to seaweed
Page 1, Line 20 and 21 – Confirm the presentation/formatting of the results
Page 1, Line 21 -The elevated reduction of iodide content of S. Latissima after heat treatment it is evident however, the authors need to indicate the conditions of treatment that reached to the reduction of 328 mg iodine kg-1 dw.
Page 1, Line 23 – eliminate the end point: “…changes were seen for ash. . A…”
Page 1, Line 25, 26 – indication of the meaning of the abbreviation observed “…the EAA/AA ratio changed…” and “…Moreover, the proportion of EPA, ALA, PUFA…”
Introduction:
Page 2, Line 49 – “…to the consumers [9]….” The authors choose the Nordic legislation for guidance of dietary values for recommended intake and upper intake level however, legislation and recommendation from the European Food Safety Authority can be found (please see the references below). In that, the tolerable upper intake level (UL) for iodine is not different that referenced by the authors: UL (ug per day) for children with 1-3 years is 200 and for children with 15-17 years is 500 and for adults is 600 ug per day.
References:
- Scientific Report of EFSA: Overview on Tolerable Upper Intake Levels as derived by the Scientific Committee on Food (SCF) and the EFSA Panel on Dietetic Products, Nutrition and Allergies (NDA)
- Scientific Committee on Food Scientific Panel on Dietetic Products, Nutrition and Allergies (2006). Tolerable upper intake levels for vitamins and minerals. European Food Safety Authority.
Page 2, Line 58-62 – Confirmation of reference number is necessary, a lack of reference number of 11 and when authors indicate the reference name and year the reference number should be added.
Page 2, Page 64-65 – For better understanding the idea of determined the nutritional value of the final product the sentence “…to investigate if the various processing conditions compromised other valuable compounds, hence the nutritional value of the final product.” Should be changed to “…to investigate if the various processing conditions compromised other valuable compounds, the nutritional value of the final product was determined.”
Page 2, Line 68 – “…ash and antioxidant capacity quantified…”, the total phenolic content should be added.
Materials and methods
Inclusion of chemicals information as brand and country of manufacture used in the present study before the point of raw material.
Page 2, Line 82 – “…JBN12 (Grant Instruments Ltd) water bath”, add the country of the company headquarter of water bath.
Page 2, Line 87 – “…-20 °C freezing room…”, The freezing was performed in the freezing room or freezer? More information is need whether the freezing was done in freezing room or freezer, for instance the dimension of freezing room and the brand of the equipment as well the country of manufacture.
Page 3, Line 93-94 – “…The rest was freeze dried at -40 °C and then homogenized by milling…” add information as indicated previously
Page 3, Line 96-98 – Add the reference for dry matter and ash determination.
Page 3 – The reference of EN 102 17050:2017 is missing.
Page 3, Line 100-114 – Missing information of centrifuge and oven… as previously commented.
Page 3, Line 121-122 – Delete the paragraph inserted.
Page 4, Line 151 – Change the title for “Extraction for antioxidant capacity and total phenolic content”
Page 4, Line 159-163 – Indicate the concentration of sodium carbonate used as well the Folin-Ciocalteau.
The authors used the propyl gallate as standard and why did not used the gallic acid an phenolic compound? The literature reported in a higher number of studies the use of gallic acid as standard in total phenolic compounds determination.
Page 4, Line 162-163 – “The measured absorbance was converted into gallic acid equivalents by a standard curve of propyl gallate in the range of 7.8 – 250 μg mL-1”. The authors used the propyl galate as standard and the results were converted into gallic acid? How this conversion was carried out?
Page 4, Line 164 – Change the title for “DPPH radical scavenging activity” and the equation (2):
DPPH radical scavenging capacity = (…)
Page 4, Line 165-166 – “…Yang, Guo and Yuan, (2008)”, add at the end of reference the corresponding number.
Results and Discussion
In Table 1 and Table 3, a few obtained means resulting from two and not three replicates as previously indicated by authors. Why did this happen? In some samples two and in others three replicates?
Page 6, Line 233-234 – “…treatments, the carbohydrates did as well and showed significant differences…” – correct the sentence
Page 3 (?), Line 330- In section 3.6 a discussion of antioxidant activity by discussion of TPC and DPPH radical scavenging activity were performed.
Page 3 (?), Line 331 – Addition of literature that was reported the antioxidant capacity and total phenolic content of fresh Saccharina latissima.
Page 3 (?), Line 340 – “…at different seasons (0.84 - 2.41 mg GAE/g…” the authors in this section presented the results in mg GAE/g however, in section of materials and methods another standard was described. The authors need to confirm the units of TPC of all samples.
In Figure 5 – The same question about the standard used and presentation of results
